# Structures of Foot-and-mouth Disease Virus with neutralizing antibodies derived from recovered natural host reveal a mechanism for cross-serotype neutralization

Yong He[1,2☯], Kun Li[3☯], Yimei Cao[3], Zixian Sun[2], Pinghua Li[3], Huifang Bao[3], Sheng Wang[3], Guoqiang Zhu[3], Xingwen Bai[3], Pu Sun[3], Xuerong Liu[4], Cheng Yang[1], Zaixin Liu[3]*, Zengjun Lu[3]*, Zihe Rao[1,2]*, Zhiyong Lou[2]*

1 State Key Laboratory of Medicinal Chemical Biology and Drug Discovery Center for Infectious Disease, College of Pharmacy, Nankai University, Tianjin, China, 2 MOE Key Laboratory of Protein Science & Collaborative Innovation Center of Biotherapy, School of Medicine and School of Life Sciences, Tsinghua University, Beijing, China, 3 State Key Laboratory of Veterinary Etiological Biology, National Foot-and-Mouth Diseases Reference Laboratory, Lanzhou Veterinary Research Institute, Chinese Academy of Agricultural Sciences, Lanzhou, China, 4 China Agricultural Vet Biology and Technology Co. Ltd., Lanzhou, China

☯ These authors contributed equally to this work.
* liuzaixin@caas.cn (ZL); luzengjun@caas.cn (ZL); raozh@tsinghua.edu.cn (ZR); louzy@mail.tsinghua.edu.cn (ZL)

**Data Availability Statement:** All relevant data are within the manuscript and its Supporting Information files. The cryo-EM density maps and

## Abstract

The development of a universal vaccine against foot-and-mouth disease virus (FMDV) is hindered by cross-serotype antigenic diversity and by a lack of knowledge regarding neutralization of the virus in natural hosts. In this study, we isolated serotype O-specific neutralizing antibodies (NAbs) (F145 and B77) from recovered natural bovine hosts by using the single B cell antibody isolation technique. We also identified a serotype O/A cross-reacting NAb (R50) and determined virus-NAb complex structures by cryo-electron microscopy at near-atomic resolution. F145 and B77 were shown to engage the capsid of FMDV-O near the icosahedral threefold axis, binding to the BC/HI-loop of VP2. In contrast, R50 engages the capsids of both FMDV-O and FMDV-A between the 2- and 5-fold axes and binds to the BC/EF/GH-loop of VP1 and to the GH-loop of VP3 from two adjacent protomers, revealing a previously unknown antigenic site. The cross-serotype neutralizing epitope recognized by R50 is highly conserved among serotype O/A. These findings help to elucidate FMDV neutralization by natural hosts and provide epitope information for the development of a universal vaccine for cross-serotype protection against FMDV.

## Author summary

FMDV is the causative agent of foot-and-mouth disease, one of the most contagious and economically devastating diseases of cloven-hoofed animals. The antigenic diversities of the currently known epitopes throughout FMDV serotypes and the lack of understanding of FMDV neutralization in natural hosts limit the development of a vaccine that is able to

the structures were deposited into the Electron Microscopy Data Bank (EMDB) and Protein Data Bank (PDB) with the following accession numbers: FMDV-OTi-B77, EMD-30558, PDB 7D3K; FMDV-OTi-F145, EMD-30559, PDB 7D3L; FMDV-OTi-R50, EMD-30560, PDB 7D3M; and FMDV-AWH-R50, EMD-30565, PDB 7D3R.

**Funding:** This work was supported by the National Program on Key Research Project of China [(2020YFA0707500, to Z.L.), (2017YFC0840300, to Z.R.), (2016YFD0501500, to Z. Liu)] and the National Natural Science Foundation of China (NSFC) [(31902288, to K.L.), (32072873, to Y.C.)]. The funders had no role in study design, data collection and analysis, decision to publish, or preparation of the manuscript.

**Competing interests:** The authors have declared that no competing interests exist.

provide cross-serotype protection. In this work, we isolated FMDV serotype O-specific neutralizing antibodies (NAbs) (F145 and B77) and a serotype O/A cross-reacting NAb (R50) from recovered natural bovine hosts and determined virus-NAb complex structures by cryo-electron microscopy at near-atomic resolution. Structures of virus-NAb complex reveal F145 and B77 engage the capsid of FMDV-O near the icosahedral threefold axis. In contrast, R50 engages the capsids of both FMDV-O and FMDV-A between the 2- and 5-fold axes, revealing a previously unknown antigenic site. This is the first time to present structure details of FMDV neutralization by natural hosts. And this work also provides epitope information for the development of a universal vaccine for cross-serotype protection against FMDV.

## Introduction

Foot-and-mouth disease virus (FMDV), a small nonenveloped single-stranded positive-sense RNA virus (genus *Aphthovirus* within the family *Picornaviridae*), is the causative agent of foot-and-mouth disease (FMD). FMD is one of the most contagious and economically devastating diseases of cloven-hoofed animals in many developing regions of Asia, Africa and South America[1–3]. FMDV exists as seven immunologically distinct serotypes (O, A, C, SAT1, SAT2, SAT3 and Asia1)[4]. Among them, serotypes O and A are the most common causative agents of FMDV outbreaks globally[5,6]. Vaccination is believed to play a predominant role in the control and prevention of FMDV[7]. However, the antigenic diversities of the currently known epitopes throughout FMDV serotypes and the lack of understanding of FMDV neutralization in natural hosts limit the development of a vaccine that is able to provide cross-serotype protection[8–11].

As the major antigen, the virus capsid of FMDV comprises 60 copies each of four virus-encoded proteins, VP1, VP2, VP3 and VP4; VP1 to VP3 form most of the capsid shell, with VP4 lining the interior surface[12,13]. At present, there is limited structural information regarding how FDMV engages with neutralizing antibody (NAb). The cryo-electron microscopy (cryo-EM) structure of the FMDV-C-S8c1-SD6 Fab complex shows that the murine Fab fragment SD6 binds to the VP1 GH-loop (residues 136–147), and the GH-loop plays a key role in the interaction with integrin receptors[14,15].

As a natural host of FMDV, bovines have a different immunoglobulin (Ig) repertoire than other vertebrates, which display restricted lengths of the third heavy chain complement determining region (HCDR3), averaging 12–16 amino acids. Bovines, however, produce antibodies with HCDR3s that average ~ 26 amino acids, with an ultralong subset (10–15% of the repertoire) that contains greater than 70 amino acids[16–18]. These characteristic Ig sequences point to bovines as a promising host for producing high-affinity and broadly neutralizing antibodies. This is exemplified by the rapid elicitation of broadly neutralizing antibodies to HIV after bovine immunization; these broadly neutralizing antibodies contain ultralong HCDR3s that are responsible for their serological breadth and potency[19].

As yet, it is unclear exactly how FMDV is neutralized by natural bovine host NAbs and whether an epitope exists that can elicit cross-serotype NAbs. In a previous study, we reported the isolation of two serotype O-specific NAbs (F145 and B77) from recovered natural bovine hosts through the single B cell antibody isolation technique[20]. In the current study, we isolated a NAb (R50) that neutralizes FMDV serotypes O and A. We determined the cryo-EM structures of FMDV serotype O (O/Tibet/99) in complex with B77 and F145 scFv, as well as R50 scFv in complex with FMDV serotype O (O/Tibet/99) and with serotype A (A/WH/CHA/

09). The near atomic level details reveal a key epitope for cross-serotype neutralization and inform a structure-based rationale to design a universal FMDV vaccine that can elicit intertypic cross-reacting NAbs.

## Results

### Isolation of antibodies from bovine plasmablasts

Upon isolating two FMDV serotype O-specific antibodies (B77 and F145) from a natural bovine host[20], we aimed to isolate a cross-serotype NAb that can neutralize both serotype O and A. FMDV (O/Tibet/99) (hereafter named FMDV-OTi) and FMDV (A/WH/CHA/09) (hereafter named FMDV-AWH) were applied as baits to isolate specific memory B cells from natural bovine host blood, which displayed strong cross-reactivity with both serotype O and A. As revealed in fluorescence-activated cell sorting (FACS), CD21$^+$IgM$^-$ O_FMDV$^+$ A_FMDV$^+$ B cells are extremely rare, displaying 21 per million PBMCs. Via FACS, a total of 96 single CD21$^+$IgM$^-$ O_FMDV$^+$ A_FMDV$^+$ B cells were sorted for the amplification of the IgG BCR variable gene and the development of a FMDV serotype O and A cross-neutralizing scFv antibody (Fig 1A). Finally, 60 paired IgG clones from these cross-reactivity B cells were successfully amplified. Of these, 14 scFv antibodies displayed cross-reactivity with both FMDV serotype O and A. But, only R50 scFv antibody showed cross-neutralizing activity with both FMDV serotype O and A.

To further investigate the neutralization breadth and potency of R50 scFv antibody, we propagated, purified three representative strains of FMDV serotype O (O/Mya/98; O/HN/CHA/93; O/Tibet/99) obtained from three currently epidemic topotypes (SEA, Cathay, and ME-SA, respectively) and two representative strains of FMDV serotype A (A/WH/CHA/09; A/GDMM/CHA/2013) from the Asia topotypes SEA-97/G1 and G2 in China[21], and separately examined their binding and neutralization abilities by flow cytometry and microneutralization assay. The flow cytometry showed that R50 scFv binds to the three topotypes of FMDV serotype O and the two distinctive strains of FMDV serotype A (Fig 1B). In line with the binding results, microneutralization assay also revealed that R50 scFv cross-neutralizes these strains of FMDV serotype O and A (Fig 1C). In contrast, B77 and F145 scFv only neutralize the three representative strains of FMDV serotype O(Fig 1C), as previously reported [20].

### Overall architectures of the FMDV-NAb complexes

To clarify the mechanism of cross-serotype neutralization, we determined the structures of B77 and F145 scFv in complex with FMDV-OTi, as well as R50 scFv in complex with FMDV-OTi and FMDV-AWH (S1 Fig). The cryo-EM micrographs indicated the presence of scFv on the FMDV capsids at distinct locations (Fig 2). The final resolution of the cryo-EM reconstruction was estimated by the FSC 0.143 cutoff to be 3.90 Å for the FMDV-OTi-B77 complex, 3.68 Å for the FMDV-OTi-F145 complex, 3.94 Å for the FMDV-OTi-R50 complex and 3.49 Å for the FMDV-AWH-R50 complex (S1 Fig).

Three B77 or F145 scFv molecules bind to the FMDV-OTi capsid around an icosahedral threefold axis (Fig 2A, 2B, 2E and 2F). By contrast, R50 scFv binds to a depression between the 2- and 5-fold axes of FMDV-OTi and FMDV-AWH (Fig 2C, 2D, 2G and 2H). In all cases, the cryo-EM densities were of sufficient quality to allow the atomic modeling of FMDV capsid proteins and the NAb scFv variable loops that are responsible for virus recognition. However, the densities of the bound NAbs away from the contact surface were not sufficiently clear to trace all main chains (S2 Fig). The footprints of B77, F145 and R50 on the capsid surface were also mapped (Fig 2I–2L).

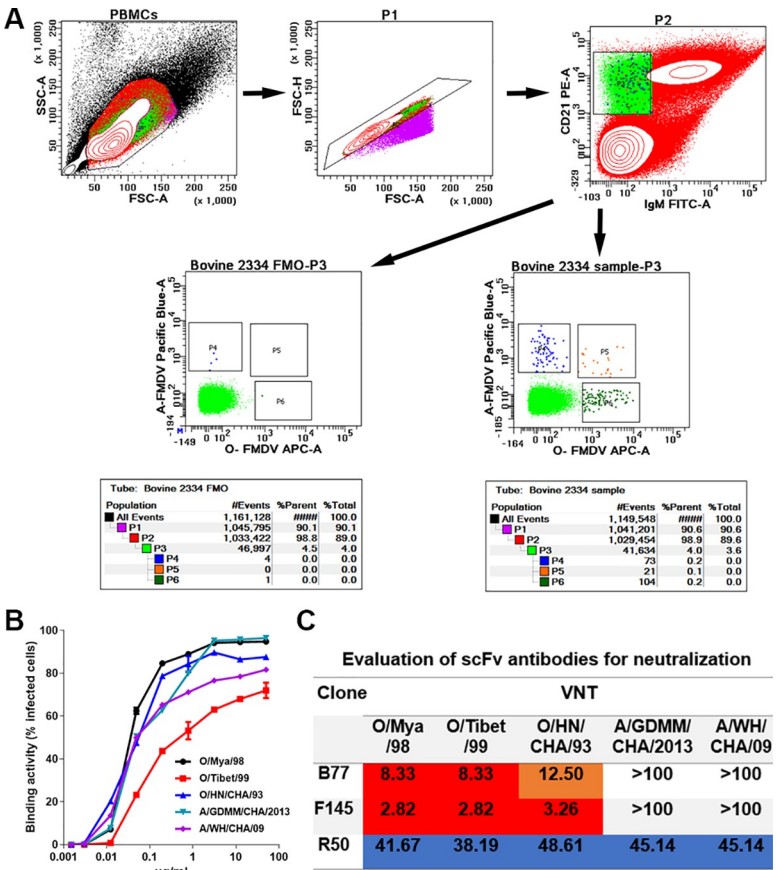

**Fig 1. Isolation of NAbs.** (A) Confirmation of the phenotype and proportion of FMDV serotype O and serotype A cross-reactive plasmablasts in the peripheral blood by flow cytometry. Bovine PBMCs were analyzed by four-color flow cytometry. Gate 1 (P1) was selected to exclude cell debris, with lower values of SSC-A and FSC-A, and was further analyzed to gate singlets (P2) based on the diagonal streak of the FSC-A vs. FSC-H plot. The CD21$^+$IgM$^-$ B cells (P3) were gated to check the distributions of O_FMDV$^-$ A_FMDV$^+$ (P4), O_FMDV$^+$ A_FMDV$^-$ (P6) and O_FMDV$^+$ A_FMDV$^+$ (P5) population, in the presence (Bovine 2334 sample-P3) or absence (Bovine 2334 FMO-P3) of bait antigens. (B)The binding activity of R50 scFv with FMDV serotype O and A strains. The O/Mya/98, O/HN/CHA/93, O/XJ/CHA/2017, A/GDMM/CHA/2013 and A/WH/CHA/09 strains were separately used to infect BHK-21 cells. The binding activity of R50-ScFv (concentration, 0–50 μg/ml) with the infected BHK-21 cells was measured by flow cytometry. Data are presented as the mean ± standard error of the mean and represent two independent experiments. (C) Evaluation of bovine-derived FMDV-neutralizing scFv antibodies for neutralization and potency. Values are neutralization titer (NT) in μg/ml. An NT value of 100 μg/ml was used as a cutoff for neutralization, and >100 μg/ml was defined as showing no virus-neutralizing activity. NT values of 0–10 μg/ml are red, 10–25 μg/ml are orange, 25–50 μg/ml are blue, and >100 μg/ml are white.

The lengths of bovine HCDR3 feature a trimodal distribution: group 1 comprises very short HCDR3s (≤10 amino acids); group 2 comprises intermediate lengths (11 to 47 amino acids); and group 3 comprises ultralong HCDR3s (≥48 amino acids)[22,23]. The structures show that B77 contains an intermediate length HCDR3 loop with 18 amino acids, but F145 and R50 contain ultralong HCDR3 loops (50 and 61 amino acids, respectively) (S3 Table). The ultralong HCDR3s form an elongated "stalk" and a globular "knob" at the distal ends of the stalks, and the stalk region is essentially a display scaffold to deliver the knob domain, which functions as an effector domain that mediates the molecular recognition of antigen. The globular knob of F145 interacts with VP2, but the globular knob of R50 binds to a shallow depression on the capsid, contacting the two major capsid proteins VP1 and VP3 from two adjacent protomers (S3 Fig).

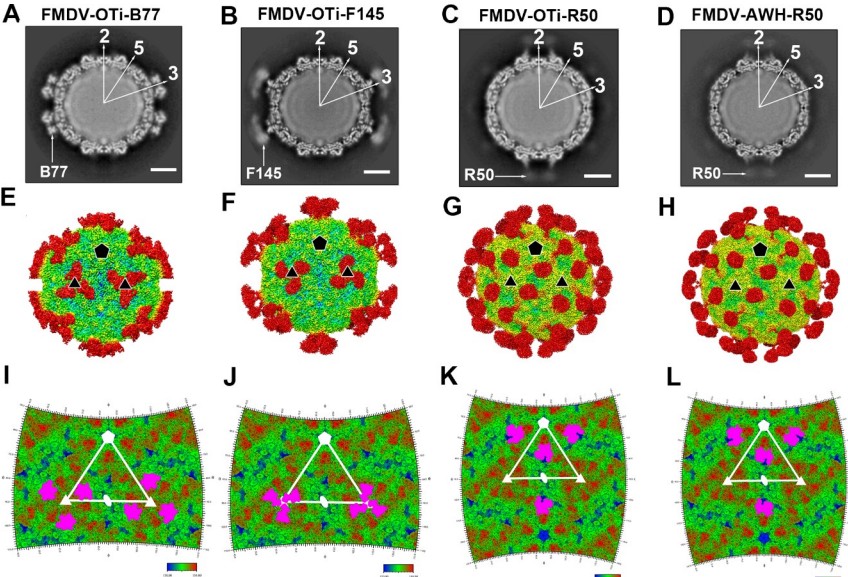

**Fig 2. Cryo-EM structures.** (A-D) The central cross-sections through the cryo-EM maps of FMDV-OTi-B77 complex (A), FMDV-OTi-F145 complex (B), FMDV-OTi-R50 complex (C) and FMDV-AWH-R50 complex (D) are shown with the icosahedral two-, three- and five-fold axes. Each image in the 480-pixel boxes corresponds to 446 Å in each dimension. Scale bars, 100 Å. (E-H) Rendered images of the FMDV-OTi-B77 complex (E), FMDV-OTi-F145 complex (F), FMDV-OTi-R50 complex (G) and FMDV-AWH-R50 complex (H). Depth cueing with color is used to indicate radius (< 120 Å, blue; 130–150 Å, from cyan to yellow; > 160 Å, red). The icosahedral five- and three-fold axes are represented by pentagons and triangles, respectively. (I-L) Footprints of B77, F145 and R50 on the FMDV surface. The figure shows a 2D projection of the FMDV surface produced using RIVEM. 5-, 3-, and 2-fold icosahedral symmetry axes are marked as pentagons, triangles, and ovals, respectively, on one icosahedral asymmetrical unit. The spherical polar angles (θ, φ) define the location on the icosahedral surface. The depictions are radially depth cued from blue (radius = 130 Å) to red (radius = 155 Å). The residues in the scFv footprints are shown in purple.

## Interaction of B77 and F145 with FMDV-OTi

The complex structures show that B77 and F145 make similar contacts with the BC-loop and HI-loop of FMDV-OTi VP2. Residues in the VP2 BC-loop ($_{OTi-VP2}$T71 and $_{OTi-VP2}$S72) and in the VP2 HI-loop ($_{OTi-VP2}$N190 and $_{OTi-VP2}$Q196) interact with residues $_{B77-VH}$M56, $_{B77-VH}$T101, $_{B77-VH}$S105, $_{B77-VH}$L108 and $_{B77-VH}$W114 in the B77 heavy chain (S4 Table). The $_{OTi-VP2}$T71 and $_{OTi-VP2}$S72 side chains form hydrogen bond contacts with side chain atoms of $_{B77-VH}$T101 and $_{B77-VH}$W114 (Fig 3A and S4 Table). The side chains of $_{OTi-VP2}$T190 and $_{OTi-VP2}$Q196 form hydrogen bond contacts with the side chain atoms of $_{B77-VH}$M56 and $_{B77-VH}$S105 (Fig 3A and S4 Table). In addition, $_{OTi-VP2}$H65 and $_{OTi-VP2}$D68 in βB and the BC-loop interact with residues $_{B77-VH}$W93 and $_{B77-VH}$S95 in the light chain of B77 (S4 Table). The side chain of $_{OTi-VP2}$H65 forms hydrogen bonds or van der Waals contacts with the side chain atoms of $_{B77-VL}$S95 (Fig 3A and S4 Table). To further confirm the epitope, neutralization escape mutants were screened under B77 immune pressure. The B77-resistant mutants were isolated and characterized. An Asn→Ser mutation occurred at position 190 of VP2; this position is located in the virus-antibody binding interface and is strictly conserved among serotype O (S4 Table and Fig 3C).

F145 interacts with the VP2 BC-loop ($_{OTi-VP2}$T71 and $_{OTi-VP2}$S72) and the HI-loop ($_{OTi-VP2}$V189, $_{OTi-VP2}$N190, $_{OTi-VP2}$T191) (S5 Table). The side chains of $_{OTi-VP2}$S72 and $_{OTi-VP2}$T191 form hydrogen bond contacts with the side chain atoms of $_{F145-VH}$S119 and $_{F145-VH}$T128(Fig 3B and S5 Table). The amide nitrogen of $_{OTi-VP2}$N190 provides additional stabilization via hydrogen bonds with the hydroxyl group of $_{F145-VH}$D129 (Fig 3B and S5 Table). To further

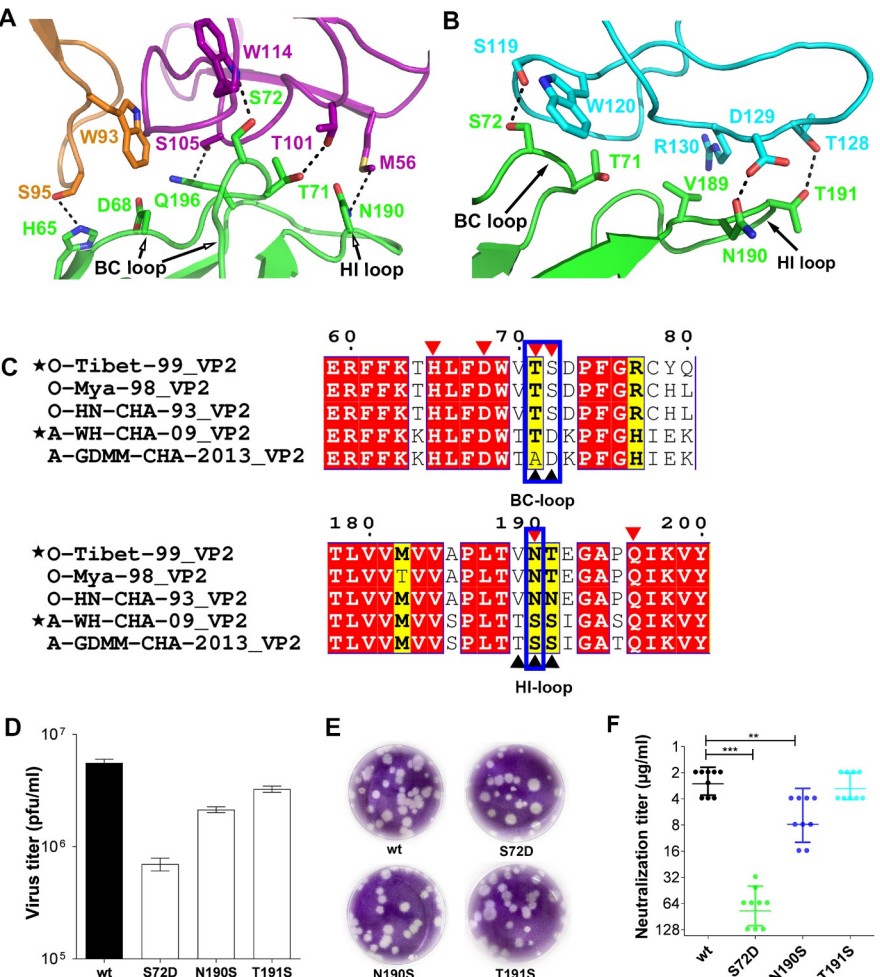

**Fig 3. B77 and F145 bind to FMDV-OTi.** (A) Close-up view of FMDV-OTi-B77 interaction interface. VH and VL of B77 are purple and orange. Contacting residues are displayed as sticks with the same color scheme as in the main chain. Dashed lines indicate contacts with potential hydrogen bonds and salt bridges. Amino acid side chains are colored blue (nitrogen) and red (oxygen). (B) Close-up views of FMDV-OTi-F145 interaction interface. VH of F145 is cyan. (C) Sequence alignment of FMDV serotype O and serotype A. Residues with red or yellow backgrounds are identical or conserved, respectively. Residues at the FMDV-OTi-B77 and FMDV-OTi-F145 interfaces are indicated with red and black triangles, respectively. The same contacting residues are highlighted in blue boxes. Alignment was performed by CLUSTALW, and the figure was generated by ESPript. (D) Virus titers (pfu/ml) of the wild-type (O/Tibet/99) and its mutants ($_{VP2}$S72D, $_{VP2}$N190S and $_{VP2}$T191S) were determined by plaque forming assay. (E) Plaque phenotypes formed in BHK-21 cells and the sizes were correlated to the CPE patterns. (F) Neutralizing efficacy of F145 against the wild-type (O/Tibet/99) and mutants ($_{VP2}$S72D, $_{VP2}$N190S and $_{VP2}$T191S) were evaluated using a micro-neutralization assay. The neutralization titer (NT) represented the lowest antibody concentration required to fully prevent CPE. *** indicates significant difference to wild-type at P<0.001. ** indicates significant difference to wild-type at P<0.01.

confirm the epitope, neutralization escape mutants were generated under F145 immune pressure. The F145-resistant mutants were isolated and characterized. A Ser→Ala mutation was found at position 72 in VP2, which is also located in the virus-antibody binding interface.

FMDV employs heparin sulfate (HS) and integrin (generally αvβ6) as receptors[24,25]. In the cellular entry of FMDV-O, HS acts as a primary receptor that allows for rapid cell binding, thereby providing more time for virus binding to integrins[26]. FMDV binding to the integrin receptor is facilitated by a conserved arginine-glycine-aspartic (RGD) motif in the exposed GH-loop of VP1[25]. Structure comparisons of FMDV-integrin and FMDV-NAbs show

obvious clashes between NAbs (B77 and F145) and the integrin receptor, suggesting that FMDV neutralization by B77 and F145 is facilitated by blocking virus-receptor interaction via steric hindrance (S7 Fig). Meanwhile, in vivo the antibodies are IgG molecules or some other multi-site entity, a situation that could result in aggregation of virus particles if one virus particle becomes crosslinked with another. Aggregation of virus particles can cause a reduction in the number of infectious virions available to interact with receptor and may result in the neutralization.

## Determinant for the serotype O-specificity of F145 and B77

F145 and B77 neutralized FMDV-OTi but did not have a neutralizing effect against FMDV-AWH (Fig 1C). The structural alignment of FMDV-OTi and FMDV-AWH capsid proteins showed no substantial conformational changes in the VP2 BC-loop and HI-loop, which are the epitopes of B77 and F145 (S4 Fig). The sequence alignment of FMDV-OTi and FMDV-AWH exhibited distinct amino acid differences on the epitopes of B77 and F145, indicating that the variation of critical amino acid residues may be the determinant for serotype-specific recognition (Fig 3C).

For B77, we identified that the neutralization escape mutation $_{OTi-VP2}N190\rightarrow_{AWH-VP2}S190$ enables the escape of antibody-mediated neutralization (virus neutralization [VN] titer >400 μg/ml) (Fig 3C and S8 Table). For F145, we mutated FMDV-OTi amino acid by FMDV-AWH's sequence at the corresponding positions (S72D, N190S and T191S), rescued virions by reverse genetics approach and evaluated the neutralization capacity using virus neutralization test (VNT). All mutated viruses could be rescued with an acceptable attenuated proliferation and cytopathic effect (CPE) (Fig 3D and 3E). The VNT results showed that FMDV-OTi with the $_{OTi-VP2}N190S$ mutation exhibited a 2.9-fold reduction in VN titer, and the $_{OTi-VP2}S72D$ mutation resulted in a significant reduction (~30-fold) in VN titer (Fig 3F). These results suggest that VP2 S72 and N190 play the most essential roles in the serotype O-specific neutralization activity of F145 and B77.

## FMDV-OTi-R50 and FMDV-AWH-R50 interfaces

R50 binds FMDV-OTi VP1 and VP3 by its heavy chain (Fig 4A). The antibody-interacting residues on the FMDV-OTi capsid are located in the VP1 BC-loop (residues $_{OTi-VP1}V50$ and $_{OTi-VP1}D52$), VP1 EF-loop (residues $_{OTi-VP1}P94$ and $_{OTi-VP1}E95$), VP1 GH-loop (residues $_{OTi-VP1}R157$, $_{OTi-VP1}L159$ and $_{OTi-VP1}P160$) and VP3 GH-loop (residues $_{OTi-VP3}D173$ and $_{OTi-VP3}T177$) (S6 Table). The interacting residues in R50 are located in HCDR3, including residues $_{R50-VH}R119$, $_{R50-VH}W120$, $_{R50-VH}Y130$, $_{R50-VH}N132$, $_{R50-VH}Y138$, $_{R50-VH}G139$, $_{R50-VH}R140$, $_{R50-VH}C141$ and $_{R50-VH}T142$ (S6 Table). The side chain of $_{R50-VH}R140$ extends into a deep cavity and forms interactions with $_{OTi-VP1}D52$ in the VP1 BC-loop and $_{OTi-VP1}L159$ in the VP1 GH-loop (Fig 4A and S6 Table). Residues $_{OTi-VP1}V50$, $_{OTi-VP1}R157$ and $_{OTi-VP1}P161$ also stabilize the virus-antibody interaction by interacting with $_{R50-VH}W120$ and $_{R50-VH}T142$ (Fig 4A and S6 Table). In addition, the small protruding structure formed by $_{OTi-VP1}P94$ and $_{OTi-VP1}E95$ of the VP1 EF-loop makes contact with $_{R50-VH}R119$, $_{R50-VH}C131$ and $_{R50-VH}N132$ (Fig 4A and S6 Table). One salt bridge between $_{OTi-VP1}E95$ and $_{R50-VH}R119$ was inferred because the centroids of the oppositely charged functional groups of the residues were within a 4 Å cutoff and because the Glu carbonyl oxygen atom was within 4 Å from the nitrogen atom of the Arg side chain (Fig 4A and S6 Table). Moreover, the side chains of $_{OTi-VP3}D173$ and $_{OTi-VP3}T177$ form hydrogen bonds with the side chains of $_{R50-VH}Y130$ and $_{R50-VH}Y138$, respectively (Fig 4A and S6 Table).

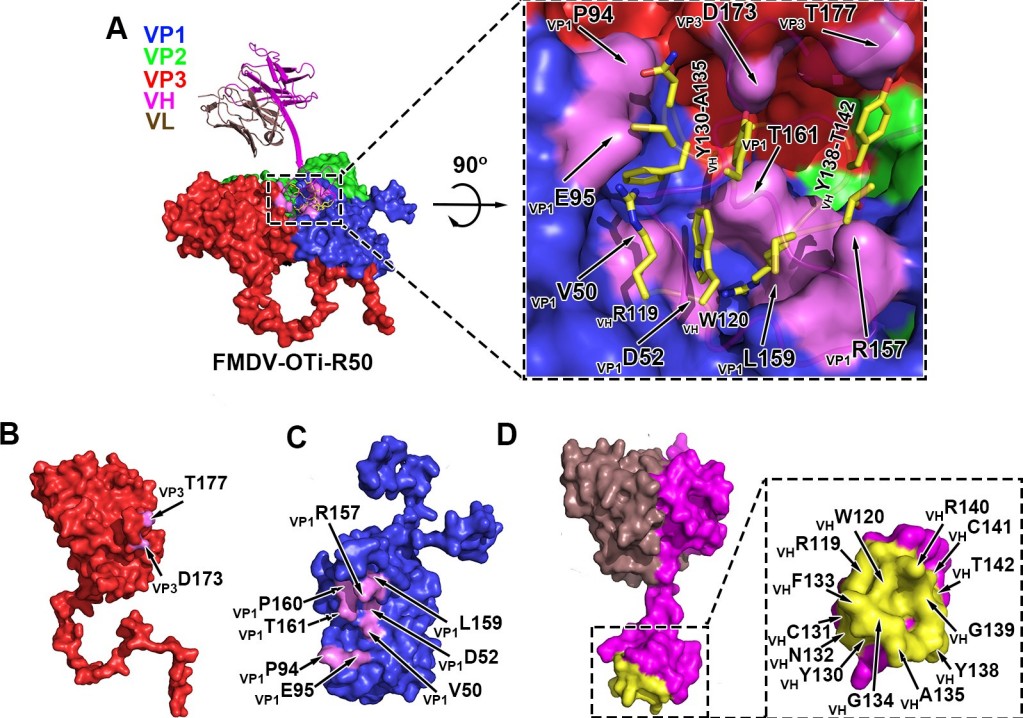

**Fig 4. Structure of FMDV-OTi-R50 complex.** (A) Surface representation of FMDV-OTi-R50 showing the interaction interface between R50 (ribbon diagram) and the capsid. The interface is framed and shown as a close-up view in the right panel. The residues in R50 responsible for interacting with the capsid are shown as yellow sticks. The interface amino acid residues in VP3 (B) and VP1 (C) are colored magenta and labeled; the interface residues in R50 are colored yellow and labeled (D).

The interaction of R50 with FMDV-AWH is homologous to that observed in the FMDV-OTi-R50 complex; R50 makes contact with FMDV-AWH VP1 and VP3 (Fig 5A). In FMDV-AWH VP1, residues that participate in antibody binding are located in the BC-loop (residues $_{AWH-VP1}$V50 and $_{AWH-VP1}$D52), EF-loop (residues $_{AWH-VP1}$P94 and $_{AWH-VP1}$E95), and GH-loop ($_{AWH-VP1}$Q156, $_{AWH-VP1}$L157 and $_{AWH-VP1}$P158) (S7 Table). The antibody-interacting residues of FMDV-AWH VP3 are located in the GH-loop (residues $_{AWH-VP3}$D174 and $_{AWH-VP3}$V175) (S7 Table). The side chain of $_{R50-VH}$R140 extends into the deep cavities and contacts $_{AWH-VP1}$D52 in the BC-loop and $_{AWH-VP1}$L157 in the GH-loop (Fig 5A and S7 Table). Residues $_{AWH-VP1}$V50, $_{AWH-VP1}$Q156, $_{AWH-VP1}$P158 and $_{R50-VH}$W120 also help stabilize the virus-antibody interaction in this region (Fig 5A and S7 Table). Additionally, the small protruding structure formed by $_{AWH-VP1}$P94 and $_{AWH-VP1}$E95 of the EF-loop contacts the residues $_{R50-VH}$R119, $_{R50-VH}$C131 and $_{R50-VH}$N132(Fig 5A and S7 Table). The side chain of $_{AWH-VP1}$E95 can form salt bridges with the side chain of $_{R50-VH}$R119. Moreover, the side chains of $_{AWH-VP3}$D174 and $_{AWH-VP3}$V175 form hydrogen bonds with the side chains of $_{R50-VH}$Y130 and $_{R50-VH}$Y138, respectively (Fig 5A and S7 Table).

The sequence alignment of FMDV serotype O and A shows that the common residues that contact R50 are strictly conserved in all five representative strains of FMDV serotypes O and A; these residues are $_{VP1}$V50, $_{VP1}$D52, $_{VP1}$P94, $_{VP1}$E95, $_{VP1}$L159 (corresponding to $_{VP1}$L157 of FMDV-AWH), $_{VP1}$P160 (corresponding to $_{VP1}$P158 of FMDV-AWH) and $_{VP3}$D173 (corresponding to $_{VP3}$D174 of FMDV-AWH) (Fig 5E). To validate the virus-antibody interactions, we substituted alanine for FMDV-OTi capsid residues whose side chains are involved in

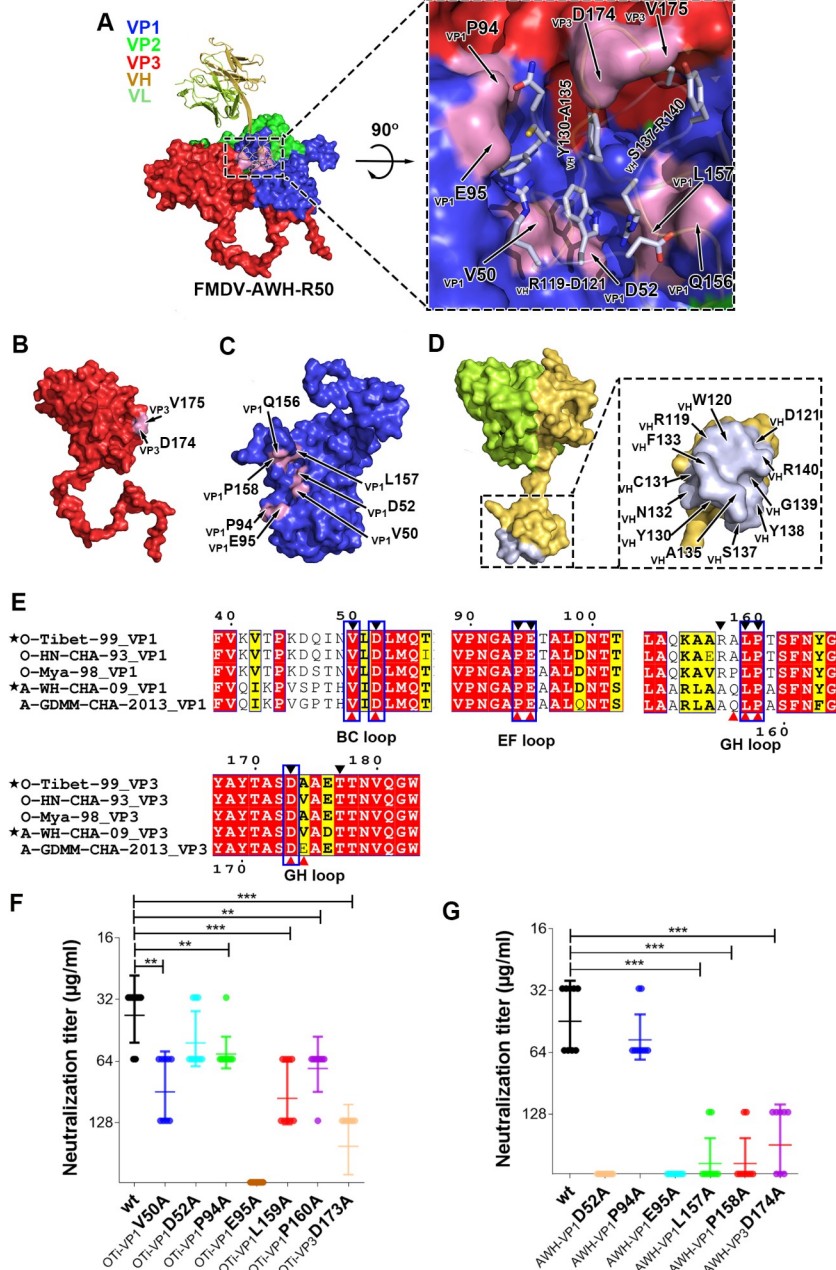

**Fig 5. Structure of FMDV-AWH-R50 complex.** (A) Surface representation of FMDV-AWH-R50 showing the interaction interface between R50 (ribbon diagram) and the capsid. The interface is framed and shown as a close-up view in the right panel. The residues in R50 responsible for interacting with the capsid are indicated by light blue sticks. The interface amino acid residues in VP3 (B) and VP1 (C) are colored pink and labeled; the interface residues in R50 are colored light blue and labeled (D). (E) Sequence alignment of FMDV type O and A. Residues with red or yellow backgrounds are identical or conserved, respectively. Residues in the FMDV-OTi-R50 and FMDV-AWH-R50 interaction interfaces are indicated with black and red triangles, respectively. The same interface residues are highlighted in blue boxes. (F) Neutralizing efficacy of R50 against the wild-type (O/Tibet/99) and mutants was evaluated using a micro-neutralization assay. The neutralization titer (NT) represented the lowest antibody concentration required to fully prevent CPE. *** indicates significant difference to wild-type at P<0.001. ** indicates significant difference to wild-type at P<0.01. (G) Neutralizing efficacy of R50 against the wild-type (A/WH/CHA/09) and mutants were evaluated using a micro-neutralization assay.

antibody binding. The mutated viruses were rescued and assessed for neutralization breadth and potency. Mutations at six individual sites ($_{OTi-VP1}$V50A, $_{OTi-VP1}$D52A, $_{OTi-VP1}$P94A, $_{OTi-VP1}$E95A, $_{OTi-VP1}$P160A and $_{OTi-VP3}$D173A) attenuated virus proliferation and produced less CPE. However, $_{OTi-VP1}$L159A increased virus proliferation and grew with greater CPE (S5 Fig). All these mutations reduced the antibody-mediated neutralization of R50. Particularly, the mutation $_{OTi-VP1}$E95A completely escaped antibody-mediated neutralization (VN titer > 250 μg/ml) (Fig 5F). Meanwhile, we also performed mutagenesis experiments on the FMDV-AWH capsid. The mutations at five individual sites ($_{AWH-VP1}$D52A, $_{AWH-VP1}$P94A, $_{AWH-VP1}$E95A, $_{AWH-VP1}$P158A and $_{AWH-VP3}$D174) attenuated virus proliferation and produced less CPE. However, $_{AWH-VP1}$L157A increased virus proliferation and grew with greater CPE (S6 Fig). All these mutations reduced the antibody-mediated neutralization of R50. In particular, the mutations $_{AWH-VP1}$D52A and $_{AWH-VP1}$E95A enabled complete escape of antibody-mediated neutralization (VN titer > 250 μg/ml) (Fig 5G). These results demonstrate the presence of conserved epitopes for the cross-serotype neutralization of FMDV by R50.

Notably, the R50 footprint on the FMDV capsid does not overlap with identified receptor binding sites (S7 and S8 Figs), indicating that R50 may not directly compete with the binding of receptor. Previous studies have reported that A9-Fab destabilized EV71 virions and as much as 70% of EV71 F-particles had been destroyed when incubated with A9-Fab at 37˚C [27]. When we incubated FMDV-OTi or FMDV-AWH with R50 at a molar ratio of 1:240 for 30 s or 60 s at 4˚C, the negative stain electron micrographs showed that R50 destabilized the virus particle and induced dissociation of virus particles into capsid pentamers, and very few particles remained intact after 60 s (S9 Fig). Meanwhile, the cryo-electron microscopy (cryo-EM) micrographs also showed significantly lower particle density when incubated for 30s (S1 Fig). In contrast to this remarkable effect, B77 did not induce significant decrease of particle density even after one hour incubation at 4˚C (S9 Fig). These results indicated a possible neutralization mechanism of R50 by promoting the dissociation of FMDV virions before infection.

## Discussion

Foot-and-mouth disease virus (FMDV) causes a highly contagious disease in bovines and other cloven-hoofed animals. Difficulties facing the prevention and control of FMDV include a high morbidity, a complex host range, and broad antigenic diversity[28]. Control of the disease in endemic regions has been mainly based on large-scale vaccinations. It is known that vaccination with one serotype cannot confer adequate protection against strains of different serotypes or heterologous strains of the same serotype based on antigenic variation; furthermore, the protection response is not immediate[29]. It is important to develop universal protective vaccines against the seven FMDV serotypes. However, the cross-serotype effectiveness of the available FMDV vaccine is limited. A previous work constructed chimeric virus-like particles displaying tandem repeats of B cell epitopes (VP1 residues 134–161 and 200–213) derived from FMDV serotypes O and A using a truncated HBc carrier; the chimeric VLPs elicited moderate neutralization antibodies against FMDV serotypes O and A in mice[30].

At present, five functionally independent neutralizing antigen sites recognized by murine monoclonal antibodies (mAbs) have been identified on the capsid surface of FMDV serotype O by neutralization escape mutants[31–34]. Site 1 is linear and trypsin-sensitive; it encompasses the GH-loop and the carboxy terminus of VP1, with critical amino acid substitutions at residues 144, 148, 154 and 208. However, sites 2–5 are all conformational and trypsin-resistant. Site 2 (isolated after pressure from C6 or C9 NAbs) is defined in the BC-loop and EF-loop of VP2 and involves critical amino acid substitutions at residues 70–73, 75, 77, and 131. Site 3 (isolated after pressure from C8 NAb) is defined in the BC-loop of VP1, with critical residues

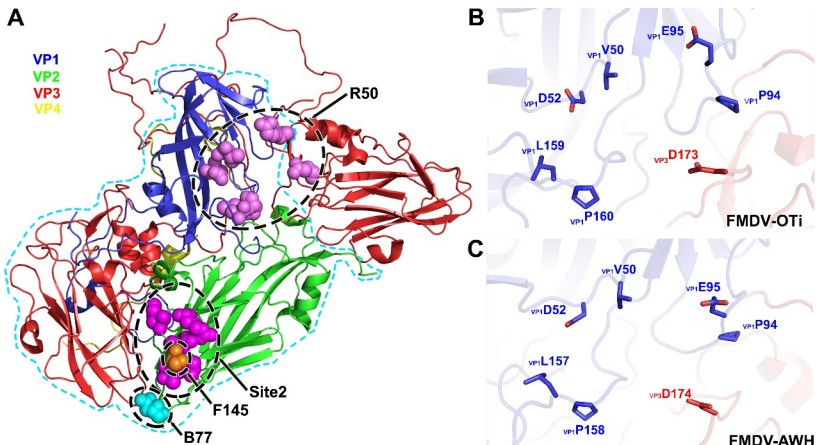

**Fig 6. Key epitope residues at the binding interface.** (A) Key epitope residues are shown as spheres on unbound FMDV capsid surface (ribbon diagram) using PyMOL. Site 2 is shown in magenta and F145 in orange. B77 is in cyan, R50 in pink. (B, C) Common residues of FMDV-OTi (B) and FMDV-AWH (C) that contact R50. Contacting residues are displayed as sticks with the same color scheme as in the main chain. In the stick models, the residue numbers are indicated. The VP1 and VP3 residues are labeled with a subscript.

at positions 43 and 44. Critical residues at position 58 in the VP3 B-B "knob" contribute to site 4 (isolated after pressure from 14EH9 NAb). Site 5 (isolated after pressure from C3 NAb) involves an amino acid change at position 149 in the GH-loop of VP1, which is distinct from site 1 even though part of the GH-loop is encompassed. Murine antibodies usually have a short HCDR3 and generally flat binding surfaces. Antigenic sites recognized by murine monoclonal antibodies are all surface-exposed and intra-protomer sites (S10 Fig). The structures of FMDV-OTi in complex with serotype O-specific NAbs F145 and B77 show that the binding sites (BC-loop and HI-loop of VP2) are also surface-exposed and intra-protomer sites. For F145, neutralization escape mutation and reverse genetics reveal that VP2 S72 within antigenic site 2 plays the most essential roles in the serotype O- specific neutralization activity (Fig 6A). For B77, the key epitope residue on VP2 N190 is the major determinant for serotype O-specific recognition. VP2-190 around the threefold axis is a new neutralizing epitope, which is close to antigenic site 2(Fig 6A).

Bovine antibodies with ultralong HCDR3 structures have the advantage to bind to epitopes that are not easily accessible to common antibodies with flatter paratope features. The structures of FMDV-OTi and FMDV-AWH with a cross-serotype NAb R50 exhibit a novel neutralizing antigenic site located in a shallow depression on the capsid, contacting two major capsid proteins VP1 and VP3 from two adjacent protomers. The novel neutralizing antigenic site formed by $_{OTi-VP1}$V50, $_{OTi-VP1}$D52, $_{OTi-VP1}$P94, $_{OTi-VP1}$E95, $_{OTi-VP1}$L159(or $_{AWH-VP1}$L157), $_{OTi-VP1}$P160 (or $_{AWH-VP1}$P158) and $_{OTi-VP3}$D173 (or $_{AWH-VP3}$D174) is highly conserved among FMDV serotypes O and A (Fig 6B and 6C). Meanwhile, NAb R50 targeting the novel antigenic site can induce a broad-spectrum cross-neutralizing response against diverse isolates of serotype O and A. Therefore, this could be a target for development of universal vaccine.

# Methods

## Ethics statement

All animal experiments in the study were approved by the Review Board of Lanzhou Veterinary Research Institute, Chinese Academy of Agricultural Sciences (permit no. LVRIAEC

2018–006) and were conducted in accordance with the Ethics Procedures and Guidelines for Animal Use of the People's Republic of China.

## Virus production and purification

FMDV (O/Tibet/99) or FMDV (A/WH/CHA/09) was grown in baby hamster kidney BHK-21 cells. Inactivated O/Tibet/99 and A/WH/CHA/09 whole virus antigens were kindly provided by an FMD inactivated vaccine manufacturer. For further purification, virus antigens in the supernatant were precipitated by incubating at 4°C overnight in 8% (w/v) PEG 6,000. The precipitated virus antigens were harvested by centrifugation at 3,500 g for 1 h at 4°C, followed by resuspension in pH 7.4 PBS (phosphate-buffered saline) (137 mM NaCl, 2.7 mM KCl, 50 mM $Na_2HPO_4$ and 10 mM $KH_2PO_4$). The viruses were purified through a sucrose cushion that transferred the samples to an ultracentrifuge tube. A 10 ml aliquot of 30% (w/v) sucrose was gently added to the bottom of the suspension, and the sample was centrifuged at 35,000 g for 1.5 h. The sucrose was removed from the pellet, and 500 µl of PBS was added to cover the pellet. The supernatant was purified over a 20–60% sucrose gradient and fractionated by centrifugation at 35,000 g for 4 h at 4°C. The fractions were analyzed by negative stain electron microscopy, and the virus fraction was transferred to a 100 kDa MWCO centrifugal filter for buffer exchange with PBS to remove the sucrose.

## Bovine infection

One 1-year-old healthy Qinchuan bovine (*Bos Taurus*), a Chinese breed of beef bovine, was raised in an animal biosafety level 3 laboratory for the sorting of antigen-specific antibody-secreting B cells after FMDV infection. The bovine, designated as #2334, was first challenged subcutaneously at two sites on the tongue with 10,000 $BID_{50}$ (50% bovine infective dose) of bovine-adapted O/Mya/98 and then received a boost vaccination with O/HN/CHA/93 and O/Tibet/99 respectively on days 35 and 132, as described in our previous report[20]. After day 225, the animal was further challenged subcutaneously at two sites on the tongue with 10,000 $BID_{50}$ (50% bovine infective dose) of FMDV (A/WH/CHA/09). Serum samples were collected at days 114 after this infection, and heparinized peripheral blood was taken from the jugular veins of the bovine for PBMC isolation.

## Identification of FMDV serotype O and A cross-neutralizing Plasmablasts in infected Bovine

PBMCs were isolated from the heparinized blood samples of the bovine with HISTOPAQUE 1.083 (Sigma-Aldrich, USA) according to the manufacturer's instructions. The PBMCs were then used to identify FMDV serotype O and serotype A cross-reactive plasmablasts. Purified FMDV (O/Tibet/99) inactivated 146S antigen was labeled with Lightning-Link Rapid FluoProbes 647H (Innova Biosciences, San Diego, USA) according to the manufacturer's instructions. Purified FMDV (A/WH/CHA/09) inactivated 146S antigen was labeled with Pacific Blue (Thermo Scientific, USA) according to the manufacturer's instructions. Freshly isolated PBMCs were stained with O-FMDV 146S-FluoProbes 647H, A-FMDV 146S-Pacific Blue, mouse anti-bovine CD21-RPE (Bio-Rad, USA) and mouse anti-bovine IgM-FITC (Bio-Rad, USA) for 30 min at 4°C in PBS containing 2 mM EDTA and 0.5% BSA. The parallel staining of PBMCs that lacked O-FMDV 146S-FluoProbes 647H and A-FMDV 146S-Pacific Blue was used as fluorescence minus one (FMO) control. The stained samples were immediately analyzed by flow cytometry, and one million PBMCs were acquired for counting the proportion of FMDV serotype O and A cross-reactive plasmablasts.

## Single-Cell sorting of FMDV serotype O and serotype A cross-neutralizing plasmablasts from bovine PBMCs using flow cytometry

The bovine PBMCs stained as above were sorted by flow cytometry (BD FACS Aria II, USA) using a 100 μm nozzle. Single-cell mode was set to sort the FMDV CD21$^+$IgM$^-$ O_FMDV$^+$ A_FMDV$^+$ cells using flow rate of 10,000 events/s. The subsequent single-cell lysis and cDNA synthesis procedures were performed in accordance with our previous report[20]. The obtained cDNA templates were stored at −20°C for subsequent PCR amplification.

## Cloning of scFv gene and construction of pcDNA3.4-scFv expression vector

Amplification of the antibody variable region genes (VH and VL) was performed by nested PCR using primers for bovine IgG gamma and lambda chains (Ig γ and Ig λ; S1 Table)[20]. Because the λ chain accounts for 95% of the bovine light chains, only the λ light chain was amplified by PCR. The final PCR products of paired VH and VL were Sanger sequenced. Single-chain fragment variable (scFv) was designed by splicing the VH and VL genes using a flexible linker (GGGGSGGGGSGGGGS). The N- and C-termini of the scFv included a signal peptide (MNPLWTLLFVLSAPRGVLS) of bovine Ig γ and a His tag, respectively. The scFv genes were synthesized by GenScript Inc. (www.genscript.com) with codon optimization and were expressed in CHO cells, followed by cloning into the pcDNA3.4 vector. The cloned scFv-expressing plasmids were amplified in *E. coli* and extracted and purified using an Endo Free Maxi plasmid kit (TIANGEN, China).

## Expression and purification of NAb scFv

Single-chain fragment variable (scFv) antibodies were expressed in ExpiCHO-S cells (Invitrogen, USA) following the manufacturer's instructions. Briefly, the CHO cells were routinely cultured at 37°C with ≥80% relative humidity and 8% CO2 in 125 mL flasks that were fixed in an orbital shaker platform at a speed of 125 rpm with a 25 mm shaking diameter. After the cell density reached $6 \times 10^6$ cells/ml, the cells were collected and resuspended in fresh media for transfection. The pcDNA3.4-scFv plasmid was transfected into ExpiCHO-S cells with an Expi-Fectamine CHO transfection kit (Invitrogen, USA). An enhancer and feed component were added at 18 h post transfection according to a standard protocol. ScFv-containing supernatants were harvested 10 days after transfection. The expressed scFv was initially purified with a HisTrap excel column. The obtained eluate was concentrated using a 100 kDa ultrafiltration tube and then further purified by size exclusion chromatography using a Superdex-200 increase 10/300 column and an AKTA plus protein purification system (GE Life Sciences, USA). The purity and size of each single-chain fragment variable (scFv) antibody was assessed by SDS-PAGE. The concentrations of the final obtained scFvs were determined by measuring their corresponding absorption values at a wavelength of 280 nm.

## Binding affinity measurement of R50 scFv

Baby hamster kidney-21 (BHK-21) cells in 24-well plate were respectively infected with different FMDV strains at a multiplicity of infection (MOI) of 0.01 and incubated for 4 h at 37°C. Subsequently, these infected cells were fixed and permeated for 20 min using a fixation and permeabilization solution (BD Biosciences, USA). After washing with PBS buffer, $1 \times 10^6$ infected cells/sample was stained with diluted R50-ScFv and subsequently incubated with anti-HIS FITC antibody (Genescript, China) and 3B4B1-Alexa647 (Fluorescence-labeled antibody (clone:3B4B1) against the nonstructural protein 3B of FMDV) for 30 min at room temperature (20–25°C). After two washes, the stained cells were loaded for FC (BD LSRFortessa, USA), and

10000 Alexa647 positive cells were recorded. The percent of FITC positive cells in Alexa647 positive cells was used to assess the binding activity of R50 scFv.

## Virus neutralizing test

The NAb scFvs were titrated for viral neutralizing activity against three representative strains from the three topotypes of FMDV serotype O and the Asia topotype of FMDV serotype A by using a microneutralization assay as previously described[21]. The five strains of FMDV were O/Mya/98 (SEA topotype), O/HN/CHA/93 (Cathay topotype), O/Tibet/99 (ME-SA topotype), A/GDMM/CHA/2013 (genotype 2 in SEA-97 lineage of Asia topotype), and A/WH/CHA/09 (genotype 1 in SEA-97 lineage of Asia topotype). Briefly, scFv antibody samples were 2-fold serially diluted in 96-well cell culture plates in a total volume of 50 µl, and 100 TCID50 of FMDV in 50 µl of culture media was added to each well. After incubation for 1 h at 37˚C, ~$5\times10^4$ BHK-21 cells in 100 µl of media were added to each well as an indicator of residual infectivity. Normal cell wells, and 10, 100, and 1,000 TCID50 virus control wells were used in each plate. The plates were incubated at 37˚C with 5% CO2 for 48 h and then fixed in acetone-methanol (volume ratio = 1:1) and stained with a 0.2% crystal violet solution. The endpoint titer was determined as the reciprocal of the lowest antibody dilution to fully prevent a cyto-pathic effect (CPE) by 100 $TCID_{50}$ FMDV in each well. The neutralizing activity was expressed as the virus neutralization (VN) titer, which was calculated as the initial antibody concentra-tion divided by the endpoint titer.

## Rescue and titration of mutated viruses

Full-length cDNAs were generated by the exchange-cassette strategy to replace the whole P1 coding region of an existing pOFS plasmid with the respective genes of O/Tibet/99 and A/WH/CHA/09. Site-directed mutagenesis was used to introduce nucleic acid mutations to pro-duce full-length cDNAs with single amino acid substitutions[35]. All mutant constructs were confirmed by nucleotide sequencing. The mutant viruses were rescued as previously described [36]. Briefly, Not I-linearized mutant plasmids were transfected into BSR/T7 cells using Lipo-fectamine 2000 following the manufacturer's instructions. The transfected cells were moni-tored daily for the appearance of CPE. At 72 h post transfection, culture supernatants were harvested and passaged on BHK-21 cells. The mutant virus titers were measured using the pla-que forming unit (PFU) assay[37]. Confluent monolayers of BHK-21 cells were infected with 10-fold serially diluted FMDV samples in 6-well plates. The 0.6% Tragacanth gum was added after 1 h of incubation. Plaques were visualized at 48 h postinfection (p.i.) by fixing with ace-tone-methanol and staining with crystal violet. The amount of plaque was observed and statis-tically analyzed.

## Generation of neutralization escape mutants

Neutralization escape mutants were generated through consecutive passages of FMDV in BHK-21 cells under the selective pressure of neutralizing scFv antibodies, according to a previ-ous report with minor modifications[31]. Briefly, tenfold serial dilutions of FMDV (O/Tibet/99) in 50 µl were incubated with 50 µl of varying concentrations of scFv sample (20 µg/ml to 50 µg/ml) in 96-well microplates. The mixtures were used to infect 100 µl of BHK-21 cells ($10^6$ cells/ml), which were incubated at 37˚C for 48 h to allow virus propagation to occur. The first passage virus was harvested from the wells seeded with the highest dilution of virus that pro-duced an approximately 80–100% cytopathic effect (CPE). Subsequent rounds of pressure selection were performed in 24-well plates in which the passaged virus (200 µl) was incubated with an equal volume of twofold concentration of antibodies in each well containing 400 µl of

BHK-21 cells. The harvested virus was subjected to several additional rounds of selection until it completely escaped from neutralization upon the addition of scFv at concentrations greater than 1 mg/ml. The P1 region of the obtained neutralization escape mutants was amplified by one-step RT-PCR as described previously[38] using the primers Pan204+: ACCTCCAACGGGTGGTACGC/NK61: GACATGTCCTCTTGCATCTG) and were subsequently verified by sequencing. Mutated amino acids were determined by aligning the entire mutant P1 region to its parent virus.

## Cryo-EM sample preparation and data collection

FMDV and antibodies were incubated at a molar ratio of 1:240 in a volume of 50 μl for 30 min (B77), 10 min (F145) and 30 s (R50) at 4˚C. A 3 μl aliquot of the mixture was applied to a glow-discharged carbon-coated gold grid (GIG, Au 1.2/1.3, 200 mesh; Jiangsu Lantuo Biotechnology Co. Ltd., China). The grid was blotted for 5 s in 100% relative humidity and plunge-frozen in liquid ethane using a Vitrobot mark IV (Thermo Fisher, USA). The cryo-EM data were collected at 200 kV with a FEI Arctica (Thermo Fisher, USA) and a direct electron detector (Falcon II, Thermo Fisher) at Tsinghua University. Micrograph images were collected as movies (19 frames, 1.2 s) and recorded at −2.4 to −1.4 μm underfocus at a calibrated magnification of ×110 kX, resulting in a pixel size of 0.932 Å per pixel. The data collection and refinement statistics are summarized in S2 Table.

## Image processing and three-dimensional reconstruction

Similar image processing procedures were employed for all data sets. Individual frames from each micrograph movie were aligned and averaged using MotionCor2[39] to produce drift-corrected images. Particles were picked and selected in Relion-2.1[40], and contrast transfer function (CTF) parameters were estimated using CTFFIND4[41] and integrated in Relion-2.1. Subsequent steps in three-dimensional (3D) reconstruction used Relion-2.1 in accordance with recommended gold-standard refinement procedures[40]. The final selected particles were further processed in THUNDER[42], which is a particle filter-based cryo-EM image processing software. The local defocus of each particle was determined by refining the defocus parameter during expectation maximization in THUNDER. For all reconstructions, the final resolution was assessed using the standard FSC = 0.143 criterion.

## Model building and refinement

The X-ray structure of native FMDV O1BFS (PDB:1BBT)[43] was manually placed into the cryo-EM map for FMDV particles and rigid-body fitted with UCSF Chimera[44]. The X-ray structure of native BOV-7 (PDB: 6E9U)[23] was manually placed into the cryo-EM map for scFv and rigid-body fitted with UCSF Chimera[44]. The fitting was further improved with real-space refinement using Phenix[45]. Manual model building was performed using Coot[46] in combination with real-space refinement with Phenix[45] to adjust the mismatches between the model and the target protein. The density maps were kept constant during the entire fitting process, and the atomic coordinates were subjected to refinement. The additional structures reported in this work were built and refined by using FMDV (O/Tibet/99) particles as a starting model and rigid-body fitted and refined. Validation was conducted using the MolProbity function integrated within Phenix. The refinement statistics are presented in S2 Table.

## Supporting information

**S1 Fig. Cryo-EM analysis of virus-NAb complex.** Typical electron micrographs were collected with a defocus of 2.0 μm (FMDV-OTi-B77), 1.9 μm (FMDV-OTi-F145), 1.5 μm (FMDV-OTi-R50) and 1.7 μm (FMDV-AWH-R50) (Scale bar, 1000 Å). Selected 2D class averages both show prominent spikes on the outer surface of viral particles (Scale bar, 480 Å). Fourier shell correlation (FSC) of the final 3D reconstruction after gold-standard refinement using RELION and THUNDER. The resolution corresponding to an FSC of 0.143 is shown for these virus-antibody complexes. FSC curves are plotted before (gray) and after (yellow) masking in addition to post correction (orange), accounting for the effect of the mask using phase randomization.
(TIF)

**S2 Fig. Cryo-EM density maps of virus-NAb complex.** Surface representation of the density maps for a protomer of these complexes. VP1, VP2, VP3 and VP4 of the protomer are blue, green, red and yellow; VH and VL of B77 are purple and orange; and VH and VL of F145 are cyan and brown, respectively. The VH and VL of R50(FMDV-OTi-R50) are magenta and dark violet, and the VH and VL of R50(FMDV-AWH-R50) are yellow orange and limon, respectively. In the right panel, atomic models shown as sticks are superimposed to indicate the representative regions in wire frames. In the stick models, the residue numbers are indicated. The VP1, VP2 and VH residues are labeled with a subscript. Black dashed circles show the densities of the bound NAbs away from the contact surface. The densities are not sufficiently clear to trace all main chains.
(TIF)

**S3 Fig. Cryo-EM structures.** Structures of the FMDV-OTi-F145 complex, FMDV-OTi-R50 complex and FMDV-AWH-R50 complex are shown in surface representation. The border of one protomer is indicated by a sky-blue line. The VP1, VP2, VP3 and VP4 of the protomer are shown in blue, green, red and yellow, while the VH and VL are shown in purple and orange, respectively.
(TIF)

**S4 Fig. Superposition of FMDV-OTi and FMDV-AWH.** Protomers of FMDV-OTi (VP1: blue, VP2: green, VP3: red) and FMDV-AWH (gray) are aligned and shown as ribbon diagram in the same orientation. The VP2 BC-loop and HI-loop are framed and shown in a close-up view in the right panel.
(TIF)

**S5 Fig. Titrations of FMDV-OTi and mutants.** (A) Titrations of rescued viruses with indicated mutations in the virus-antibody interface. The data are shown as the mean of triplicates with S.D. (B) Plaques formed in BHK-21 cells by wild-type and mutants. The patterns of CPE correlated with the plaque size.
(TIF)

**S6 Fig. Titrations of FMDV-AWH and mutants.** (A) Titrations of rescued viruses with indicated mutations in the virus-antibody interface. The data are shown as the mean of triplicates with S.D. (B) Plaques formed in BHK-21 cells by wild-type and mutants. The patterns of CPE correlated with the plaque size.
(TIF)

**S7 Fig. Binding modes of FMDV receptor and antibody.** Binding modes of receptor [integrin (avβ6) and heparin sulfate (HS)] and scFv antibody B77 (A), F145 (B) and R50 (C, D). The

panel shows a view down onto the capsid surface. VP1, VP2, VP3 and VP4 of the protomer are blue, green, red and yellow, respectively. The integrin (avβ6) and antibodies (B77, F145 and R50) are drawn in ribbon diagram; integrin (avβ6) is orange and antibodies (B77, F145 and R50) are purple. The heparin sulfate (HS) is drawn in cyan stick representation. Black dashed circles show significant clashes between antibody (B77 and F145) and integrin receptor.
(TIF)

**S8 Fig. Footprint of R50 and receptor on the FMDV surface.** Residues identified for R50 are indicated in magenta, and residues identified for heparin sulfate (HS) and integrin (generally avβ6) are indicated in cyan and orange, respectively. The border of one protomer is indicated by a yellow line.
(TIF)

**S9 Fig. Negative stain analysis.** (A, B) Negative stain EM analysis of FMDV-OTi (A) or FMDV-AWH particles when incubated with R50 at 4˚C for 30s or 60s. Red circle: pentamer, top view. Yellow circle: pentamer, side view. (C) Negative stain EM analysis of FMDV-OTi particles when incubated with B77 at 4˚C for 30 min or 60 min.
(TIF)

**S10 Fig. Antigen sites recognized by mouse monoclonal antibodies and R50.** Sites 1–5 identified for murine monoclonal antibodies are indicated in magenta; residues identified for R50 are indicated in pink.
(TIF)

**S1 Table. Nested PCR primers used for amplifying variable regions of cattle IgG.**
(DOCX)

**S2 Table. Cryo-EM data collection and refinement statistics.**
(DOCX)

**S3 Table. HCDR3 amino acid sequences of bovine antibodies.**
(DOCX)

**S4 Table. FMDV-OTi-B77 interaction residues.**
(DOCX)

**S5 Table. FMDV-OTi-F145 interaction residues.**
(DOCX)

**S6 Table. FMDV-OTi-R50 interaction residues.**
(DOCX)

**S7 Table. FMDV-AWH-R50 interaction residues.**
(DOCX)

**S8 Table. Selection of neutralization-resistant FMDV.**
(DOCX)

## Acknowledgments

We thank the Computing and Cryo-EM Platforms of Tsinghua University, Branch of the National Center for Protein Sciences (Beijing) for providing facilities.

## Author Contributions

**Conceptualization:** Yong He, Kun Li, Zengjun Lu, Zhiyong Lou.

**Formal analysis:** Yong He, Kun Li, Zixian Sun.

**Funding acquisition:** Kun Li, Yimei Cao, Zaixin Liu, Zihe Rao, Zhiyong Lou.

**Investigation:** Yong He, Kun Li, Yimei Cao, Pinghua Li, Huifang Bao, Sheng Wang, Guoqiang Zhu, Xingwen Bai, Pu Sun.

**Resources:** Xuerong Liu.

**Supervision:** Cheng Yang, Zengjun Lu, Zihe Rao, Zhiyong Lou.

**Validation:** Yong He, Kun Li, Zhiyong Lou.

**Writing – original draft:** Yong He, Kun Li, Zihe Rao, Zhiyong Lou.

**Writing – review & editing:** Yong He, Kun Li, Zhiyong Lou.

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
