## [Decision Letter · Decision Letter 0]

28 Jan 2021

Dear Dr. Lou,

Thank you very much for submitting your manuscript "Structures of Foot-and-mouth Disease Virus with neutralizing antibodies derived from recovered natural host reveal a mechanism for cross-serotype neutralization" for consideration at PLOS Pathogens. As with all papers reviewed by the journal, your manuscript was reviewed by members of the editorial board and by several independent reviewers. In light of the reviews (below this email), we would like to invite the resubmission of a significantly-revised version that takes into account the reviewers' comments.

Please respond to all of the reviewers' comments. These will be evaluated and may be resent to the reviewers so do address all concerns.

We cannot make any decision about publication until we have seen the revised manuscript and your response to the reviewers' comments. Your revised manuscript is also likely to be sent to reviewers for further evaluation.

Sincerely,

Richard J. Kuhn, PhD

Associate Editor

PLOS Pathogens

Guangxiang Luo

Section Editor

PLOS Pathogens

Kasturi Haldar

Editor-in-Chief

PLOS Pathogens

orcid.org/0000-0001-5065-158X

Michael Malim

Editor-in-Chief

PLOS Pathogens

orcid.org/0000-0002-7699-2064

Please respond to all of the reviewers' comments. These will be evaluated and may be resent to the reviewers so do address all concerns.

Reviewer's Responses to Questions

**Part I - Summary**

Reviewer #1: In this study the authors take two previously identified antibodies that are specific for binding to the FMDV O serotype. They also identify one antibody that binds both O and A serotypes. The authors express the antibody, anddo CryoEM on all three binding antibodies to there respective serotype. The authors then find some mutants in the area by co-incubating with antibody. This report has some nice Cryo-EM data, and nice mapping of antibody-capsid interactions. However I am not convinced that this area will/could be used for a universal FMDV vaccine. Particularly without real neutralization data showing cross reactivity between a diverse set of within serotype(s) and comparing other serotypes as well, Asia, C, etc.

1. The authors do not show any neutralization assay for the new cross-reactive antibody to any virus. Levels of cross neutralization between multiple isolates in the 0 and A serotypes should be included. A binding antibody does not mean neutralization. The level of neutralization is also important. Does this cross reactive antibody neutralize 0 serotypes at the same levels as the two previously identified antibody's? Does it neutralize the A serotype at the same level?

2. In the search for a universal FMDV vaccine as the authors describe, it is important that the side of binding is highly conserved among within serotype and between serotypes. Is the identified area conserved, and how high affinity does this antibody bind? It would be good to see a isolate that doesn't have this region conserved as a control.

3. The authors examine mutants in the ab-interface, most do not show a real decrease, with only a couple showing one log decrease. This is interesting, how do these mutants change binding affinity or neutralization using the new antibody?

4. It would be interesting if this new antibody could be tested in an animal experiment to show antibody treatment can prevent infection for both A and 0 serotypes.

Reviewer #2: Structures of Foot-and-mouth Disease Virus with neutralizing antibodies derived from recovered natural host reveal a mechanism for cross-serotype neutralization

This study describes the first structures of FMDV-antibody interactions using mAbs isolated from a natural bovine host and shows the potential importance of the long HCDR3 in bovine antibodies against FMDV. One of the mAbs (isolated after sequential infection with two different serotypes of the virus) is able to neutralise both of these serotypes. This is a breakthrough in the field of FMDV (cross-reactive mAbs are usually non-neutralising) and may be due to the long HCDR3 allowing binding into a depression which may not be so accessible to the murine mAbs historically used to map the antigenicity of FMDV. This is novel and important information for both FMDV and for bovine antibodies which is comprehensively described and generally well supported by the data.

Generally well written but parts of this paper are difficult to follow. Several parts of the story require reference to the 20 items of supplementary material, why can’t more of these be included in the main body? The rationale and flow for how the approaches are linked could be clearer. Subheadings which provide a description/take home of the main findings would be helpful in guiding the reader through the story.

Reviewer #3: (No Response)

**Part II – Major Issues: Key Experiments Required for Acceptance**

Reviewer #1: see above.

Reviewer #2: R50 induced virus dissociation: What are the control conditions, I think the controls for this should include other neutralising mAb/scFv which presumably will not induce dissociation? See additional comments on this in the minor issues section.

Reviewer #3: (No Response)

**Part III – Minor Issues: Editorial and Data Presentation Modifications**

Reviewer #1: none

Reviewer #2: Introduction

It is suggested that bovine antibodies with ultralong HCDR3 may be better for producing high-affinity and broadly neutralizing antibodies. I understand this was the case for bovine antibodies identified to have broadly neutralising activity against HIV but what is the general basis for the advantage? Can an antibody with longer HCDR3 make contact with a conserved epitope comprising more distant amino acids, or perhaps tolerate more variation in the positioning of components of a conserved epitope?

It is proposed that this study will 'inform a structure-based rationale to design a universal FMDV vaccine'. I wondered if the authors would like to elaborate on this rationale, could this be mentioned in the discussion?

Isolation of antibodies from bovine plasmablasts

Fig 1A I don’t understand the gating strategy based on the text in the results section and the figure legend. Please can this be explained more clearly? For example, the schematic in the figure splits into ‘FMO’ and ‘sample’ what are these? It would also be interesting to provide brief summary in the main text of how the animals were infected, how many B cells were sorted, what proportion were FMDV-specific and how many FMDV reactive B cells were characterised in order to find the three mAbs described. Can this information be extrapolated to discuss the proportion of cross-reactive antibodies that could be expected in the polyclonal response? How do the neutralising titres of the three scFv compare with the range of titres generally expected for FMDV reactive mAbs?

Overall architectures of the FMDV-NAb complexes

Three copies of B77 and F145 clearly bind to sites located closely around the 3-fold axis. R50 is described as binding near the quasi-5-fold axis, but it doesn’t seem to me to be particularly near to the 5-fold or really any symmetry axis. If anything, the closest feature seems to be the pentamer-pentamer boundary across the 2-fold and this is also where the signal is seen in the maps. So I don’t really understand why the R50 binding site is described as near the 5-fold and chosen to be displayed as binding round the 5-fold. Perhaps this should be explained?

Interaction of B77 and F145 with FMDV-OTi

Neutralisation is suggested as being by steric hindrance to integrin receptor binding. Can the authors comment on the receptor usage of these viruses, they are grown in BHK cells suggesting they are cell culture adapted to use heparan sulphate so may not require integrins, would this change the interpretation?

Determinant for the serotype O-specificity of F145 and B77

Lines 209-219: Apologies but I could not understand all of this section. The rationale for this section is not clear. Are you creating new ‘artificial escape mutant viruses’ here by reverse genetics? The text says into FMDV-AWH but then in the rest of the section only the O virus is described… I think this section needs to be made more clear.

FMDV-OTi-R50 and FMDV-AWH-R50 interfaces

Figures 4 and 5 are beautiful but if the orientation of the structure must be changed between the parts of the figure it would help if this was pointed out (e.g. the view of the enlarged inset panels is rotated).

R50 induced virus dissociation: how does the molar ratio of R50 used for dissociation compare with the concentrations of R50 used in the neutralisation experiments? What are the control conditions, I think the controls for this should include other neutralising mAb/scFv which presumably will not induce dissociation? The EM images are not high enough magnification to see the presence of the highlighted pentamers, could an enlarged inset be used to convey this point? Not a requirement but sucrose density gradient analysis would provide more positive evidence for dissociation into pentamers. A few words to introduce the dissociation of FMDV would also be useful and perhaps discussion somewhere of other mAbs with similar properties against FMDV or comparable destabilisation of other picornaviruses would also be useful.

Fig S5 seems to have a problem with poor quality plaque assay images.

Reviewer #3: (No Response)

PLOS authors have the option to publish the peer review history of their article (what does this mean?). If published, this will include your full peer review and any attached files.

Reviewer #1: No

Reviewer #2: No

Reviewer #3: No
---

## [Editor Report · Decision Letter 1]

25 Mar 2021

Dear Dr. Lou,

We are pleased to inform you that your manuscript 'Structures of Foot-and-mouth Disease Virus with neutralizing antibodies derived from recovered natural host reveal a mechanism for cross-serotype neutralization' has been provisionally accepted for publication in PLOS Pathogens.

Best regards,

Richard J. Kuhn, PhD

Associate Editor

PLOS Pathogens

Guangxiang Luo

Section Editor

PLOS Pathogens

Kasturi Haldar

Editor-in-Chief

PLOS Pathogens

orcid.org/0000-0001-5065-158X

Michael Malim

Editor-in-Chief

PLOS Pathogens

orcid.org/0000-0002-7699-2064

The authors have responded to all queries and have made substantial improvements to the manuscript.
---

## [Editor Report · Acceptance letter]

14 Apr 2021

Dear Dr. Lou,

We are delighted to inform you that your manuscript, "Structures of Foot-and-mouth Disease Virus with neutralizing antibodies derived from recovered natural host reveal a mechanism for cross-serotype neutralization," has been formally accepted for publication in PLOS Pathogens.

Best regards,

Kasturi Haldar

Editor-in-Chief

PLOS Pathogens

orcid.org/0000-0001-5065-158X

Michael Malim

Editor-in-Chief

PLOS Pathogens

orcid.org/0000-0002-7699-2064